# Dynamic Motion-Based Optimization of Support and Transmission Mechanisms for Legged Robots

**DOI:** 10.3390/biomimetics10030173

**Published:** 2025-03-11

**Authors:** Kun Zhang, Zhaoyang Cai, Lei Zhang

**Affiliations:** School of Intelligent Science and Technology, Beijing University of Civil Engineering and Architecture, Beijing 102616, China; 2108550022018@stu.bucea.edu.cn (K.Z.); caizhaoyang@bucea.edu.cn (Z.C.)

**Keywords:** legged robot, mechanism design, parameter optimization, dynamic motion

## Abstract

In order to improve the dynamic performance of legged robots, this paper proposes a method for optimizing the parameters of the leg mechanism based on dynamic motion. The proposed method consists of two key parts as follows: support mechanism optimization and transmission mechanism optimization. For the support mechanism, a mechanism analysis index based on robot motion energy is introduced to evaluate the robot dynamic motion performance. Under the structure stiffness constraint, this index can quantitatively analyze the influence of the range of motion and structure mass on the robot motion performance, thereby guiding the design of parameters such as the range of motion, structure thickness, and U-flange position of the mechanism. For the transmission mechanism, this paper optimizes the linkage length and knee joint angle for transmission ratio. Considering the variable transmission ratio and robot motion characteristics, the parameters are optimized to reduce the torque and speed requirements of the leg joint. This method determines the optimal mechanism parameters for dynamic performance based on the specified motion energy requirements, and it also optimizes the linkage length. The results show that the peak torque of the knee joint motor is reduced by 18.5%, and the peak speed is reduced by 24.8%.

## 1. Introduction

Legged robots possess a significant advantage in adapting to complex environments [1,2,3,4]. The leg support and transmission mechanisms directly influence the robot dynamic motion capabilities. Researchers propose various methods focusing on both parts, such as optimizing transmission efficiency to improve energy utilization and refining structure design to reduce leg mass [5,6]. These advancements have significantly enhanced robot dynamic performance.

For support mechanism optimization, by appropriately setting the mechanism parameters of the leg, it is possible to reduce leg inertia [7]. Sha et al. [8] proposed a lightweight design method through finite element analysis to achieve the weight reduction of robots. Ding et al. [9] established a dynamic model of the transmission system and optimized the leg mechanism parameters with the goal of minimizing the total leg mass. Vazquez-Santacruz et al. [10] introduced a novel mechatronic design method that integrates the V-model for electromechanical design, genetic algorithms, and topology optimization, leading to the development of a large-scale biped robot system. Haberland and Kim [11] have demonstrated that the bending direction of the knee joint in legged robots significantly affects motion efficiency, providing a theoretical foundation for designing more energy-efficient leg mechanisms. Sun [12] presented a topology-optimization-based method to achieve the efficient design of compliant robotic legs. Ananthanarayanan et al. [13] proposed a novel leg transmission mechanism and developed an optimization strategy that balances torque and speed by reducing leg inertia while ensuring the leg can withstand substantial load forces, thereby effectively lowering motor power consumption. However, these studies fail to consider the interrelationship between range of motion, structure stiffness, and dynamic characteristics.

For the transmission mechanism, researchers improve the dynamic performance of the structure by adjusting the transmission mode or optimizing the structure parameters. Traditional transmission mechanisms, such as harmonic transmission and belt transmission [14,15], typically mount the actuator at the knee joint directly. While this design simplifies the drive system, it significantly increases the leg inertia and also raises the torque requirements for the upper joints [16]. To address this issue, several innovative mechanism designs have emerged. For example, Pandora employs a ball screw for knee joint actuation [17], while Cassie uses a linkage system to elevate the knee actuator closer to the hip [18]. Walk-Man incorporates a single-stage transmission, driving knee pitch through a four-bar linkage, relocating the knee actuator to the thigh [19]. Meng et al. [20] developed a motor actuator with explosive output, enhancing the dynamic performance of the robot joints. Dong et al. [21] proposed using a ball screw structure and a motion trajectory optimization method to improve the dynamic performance of the legs. Michael Chadwick [22] uses the single rigid body dynamics trajectory optimization tool, TOWR, to generate realistic motion plans. Then, the predefined planned forces and motions are used to identify actuator velocities and torques. Next, the leg design parameters are optimized using a genetic algorithm. Although these optimization methods for high-dynamic leg structures have been developed for transmission mechanism, the optimization of the calf joint angle and link length parameters when using a four-bar linkage transmission has not been considered.

This paper proposes a parameter optimization method covering support mechanism optimization and transmission mechanism optimization. These two optimization processes complement each other and are both aimed at improving the dynamic performance of the legged robot. When the support mechanism parameters are fixed, the length of the fixed link in the four-bar linkage can be known, thereby guiding the design of the transmission mechanism. For the parameter optimization of the support mechanism, we propose a mechanism analysis index based on robot motion energy to evaluate the dynamic performance of robot motion. This index, under the premise of satisfying stiffness constraints, enables a quantitative analysis of the impact of the motion range and structure mass on robot performance, providing a theoretical foundation for optimal design. On the one hand, we model the equivalent U-shaped flange at the lower end of the leg mechanism as a cantilever beam structure. By conducting a mechanical analysis of the leg and an internal force analysis of the supporting mechanism, we establish the relationship between the U-shaped flange position and overall mass. On the other hand, we analyze the motion trajectory of the transmission mechanism to derive the relationship between the U-shaped flange and the knee joint range of motion, which helps determine the feasible design space for the flange. Ultimately, through the analysis of the interdependencies among these factors, the design parameters for the supporting mechanism are established. In the parameter optimization of the transmission mechanism, a kinematic analysis of the linkage is conducted to derive the relationship between the length of the linkage and the transmission ratio. The corresponding reduction ratio curve is then obtained by varying the linkage lengths. Subsequently, an optimal control method is employed to generate the joint motion characteristic curve during robot movement, which is divided into high-speed and high-torque regions. Finally, the optimal transmission ratio curve is determined by comparing the joint characteristic curves, leading to improved peak torque and peak speed output performance for the knee joint, thereby enhancing the dynamic performance of the leg structure.

The proposed method determines the parameters of the leg mechanism and provides effective guidance for designing a leg mechanism with high dynamic performance. The structure of this paper is organized as follows: Section 2 details the mechanical design of the leg. Section 3 examines the characteristics of the support mechanism and discusses the selection of its design parameters. Section 4 focuses on optimizing the linkage parameters and the shank angle to address the trade-off between speed and torque. Section 5 introduces the simulation experiments of the article. Section 6 describes the conclusion of the article. The overall framework is illustrated in Figure 1.

## 2. Analysis of the Mechanical Mechanism

### 2.1. Design Goals

The design of the leg mechanism plays a critical role in shaping the motion capabilities of legged robots. The design process involves hte following three key stages: support mechanism design, transmission mechanism design, and parameter optimization. To address the high dynamic behavior of the motion system, the primary objectives driving the optimization of the leg design are as follows:
Reduce the torque and speed requirements of the joints.High-stiffness leg mechanism.Low-inertia leg mechanism.

### 2.2. Design of the Support Mechanism

The leg support mechanism is positioned between the thigh and the knee joint. One end is fixed to the thigh joint motor, while the other supports the knee joint, enabling its rotational movement. This mechanism ensures a stable connection between the thigh and the knee while withstanding the forces and torques generated during motion. Additionally, it provides essential support for the knee flexible movement. The design of the support mechanism is critical for enhancing the robot’s leg mobility and stability.

The design of the support plate mechanism needs to prioritize the ease of installation of the knee joint motor. It should also ensure there is sufficient space for the motion of the four-bar linkage. The motor is mounted within the circular region above the support mechanism, a placement that minimizes the leg mechanism inertia and enhances overall system performance. Support flanges are evenly distributed between the two support plates, functioning as both connectors and structure reinforcements. The knee joint is positioned at the circular opening at the lower part of the mechanism, where a bearing is installed to enable smooth hinged motion between the four-bar linkage and the knee joint. As shown in Figure 2. The detailed parameters of the support mechanism are introduced in Section 3.

### 2.3. Leg Transmission Mechanism Design

The leg mechanism employs a series drive configuration, in which the motor driving the knee joint is coaxially aligned with the hip joint motor instead of being directly positioned at the knee. It reduces the additional inertia imposed by the knee motor on the hip joint [23], as shown in Figure 3.

The knee joint is driven by a four-bar linkage mechanism, which offers a variable transmission ratio to improve the robot performance. To conserve space in the leg, the motor is directly connected to the linkage mechanism. The initial parameters of the four-bar linkage components are determined based on the key mechanical requirements, range of motion, and assembly feasibility. These parameters are iteratively refined during the optimization process to enhance performance.

### 2.4. Four-Link Kinematics

A four-bar linkage mechanism is employed to actuate the knee joint motion, as depicted in Figure 4a. This mechanism offers benefits due to its simple mechanism and compact design, making it ideal for applications in confined spaces. Moreover, the four-bar linkage exhibits a variable, where adjusting the lengths of the linkages can enhance torque transmission, thereby increasing operational efficiency [24]. Kinematic analysis of this linkage is performed to inform further optimization efforts.

The solution can be derived through geometric methods. By introducing the auxiliary line AB, as illustrated in Figure 4b, the following relationships are established:(1)$$l_{AB}^{2}=l_{1}^{2}+l_{4}^{2}-2l_{1}l_{4}\cos\theta_{1}$$(2)$$\varphi_{1}=\arcsin\left(\frac{l_{1}}{l_{AB}}\sin\theta_{1}\right)$$(3)$$\varphi_{2}=\arccos\left(\frac{l_{AB}^{2}+l_{3}^{2}-l_{2}^{2}}{2l_{AB}l_{3}}\right)$$(4)$$\theta_{3}=\pi -\varphi_{1}-\varphi_{2}$$(5)$$\theta_{2}=\arcsin\left(\frac{l_{3}\sin\theta_{3}-l_{1}\sin\theta_{1}}{l_{2}}\right)$$

The closed vector equation for the four-bar mechanism can be formulated to represent the position vectors of each link as follows:(6)$$\vec{l_{1}}+\vec{l_{2}}=\vec{l_{3}}+\vec{l_{4}}$$

Its complex form can be expressed as follows:(7)$$l_{1}e^{i\theta_{1}}+l_{2}e^{i\theta_{2}}=l_{3}e^{i\theta_{3}}+l_{4}$$

By taking the derivative of the above equation with respect to time *t*, we obtain the relationship for the velocity as follows:(8)$$l_{1}\omega_{1}e^{i\theta_{1}}+l_{2}\omega_{2}e^{i\theta_{2}}=l_{3}\omega_{3}e^{i\theta_{3}}$$

By separating the real and imaginary parts of the above equation, we obtain the following:(9)$$l_{1}\omega_{1}\cos\theta_{1}+l_{2}\omega_{2}\cos\theta_{2}=l_{3}\omega_{3}\cos\theta_{3}$$(10)$$l_{1}\omega_{1}\sin\theta_{1}+l_{2}\omega_{2}\sin\theta_{2}=l_{3}\omega_{3}\sin\theta_{3}$$

The above equation can be expressed in matrix form as follows:(11)$$\left[\begin{array}{cc} -l_{2}\sin\theta_{2} & l_{3}\sin\theta_{3}\\ l_{2}\cos\theta_{2} & -l_{3}\cos\theta_{3} \end{array}\right]\left[\begin{array}{c} \omega_{2}\\ \omega_{3} \end{array}\right]=\omega_{1}\left[\begin{array}{c} l_{1}\sin\theta_{1}\\ -l_{1}\cos\theta_{1} \end{array}\right]$$

Solving the above equation yields the angular velocity $\omega_{3}$ of link $l_{3}$. The meanings of these symbols are depicted in Figure 4.

## 3. Support Mechanisms Optimization

The leg support structure is a key component in bearing loads during robot motion. The U-shaped flange, as a cantilever structure, typically requires an increase in thickness to ensure structure stiffness. However, excessive thickening may lead to an increase in mass, which in turn raises leg inertia and affects the robot’s motion performance. Therefore, it is crucial to perform a mechanical analysis and optimization of this structure.

### 3.1. Force Analysis of the Leg

To enable the robot to perform high-intensity tasks, a mechanical analysis of the robot’s jumping motion is conducted. Specifically, the analysis considers the scenario where the robot reaches a certain jump height and lands on both feet. Based on the following relationships:(12)$$H=\frac{1}{2}{gt}_{d}^{2}\;\;\;\;\;\;\;\;\;\;V_{m}=gt_{d}\;\;\;\;\;\;\;\;\;\;Ft_{m}=mV_{m}$$
and the following can be concluded:(13)$$F=\frac{m\sqrt{2gH}}{2t_{m}}$$
where *X* is the jump height, *g* is the gravitational acceleration, $t_{d}$ is the robot fall time, $t_{m}$ is the time when the robot feet make contact with the ground and the velocity becomes zero, $V_{m}$ is the instantaneous velocity upon ground contact, and *F* represents the force exerted by the ground on the robot feet during the fall. The force *F* can be decomposed into $F_{1}$ and $F_{2}$, as shown in Figure 5, resulting in the following:(14)$$F_{1}=F\cos\theta_{a}$$

When the robot foot is subjected to a lateral force $F_{1}$, in order to illustrate the internal forces at the knee joint acting on the support mechanism under this external force, the leg mechanism can be treated as a whole. A plane, denoted as m-m, divides the leg mechanism into the following two parts: I and II. Under the action of the external force *F*, to maintain force equilibrium in part II, part I must exert a force on the m-m section of part II to balance the external force applied to part II, as shown in Figure 6. The external force *F* causes part II to displace along the x-axis and rotate about point *O*. Consequently, part I must exert an internal force $F_{N}$ and a moment *M* on the section to ensure the equilibrium of part II. Here, $F_{N}$ represents the shear force on the section passing through point *O*, and *M* represents the moment about point *O*.

By applying the following equilibrium conditions:(15)$$\sum F_{X}=0\;\;\;\;\;\;F-F_{N}=0$$(16)$$\sum M_{O}=0\;\;\;\;\;\;Fa-M=0$$
the internal force $F_{N}$ and moment *M* are determined as follows:(17)$$F_{N}=F_{1}\;\;\;\;\;\;M=Fa$$

From the above discussion, it can be concluded that when a lateral force $F_{N}$ is applied to the robot foot, the support mechanism must withstand the lateral force $F_{N}$.

### 3.2. Mechanical Performance Analysis of the Support Mechanism

By analyzing the forces acting on the leg, the force $F_{N}$ exerted on the leg support plate is determined. Based on this, the mechanical performance of the support mechanism is further analyzed. For the purposes of our study, the model of the support mechanism is simplified, and its mechanical performance is subsequently computed. To analyze the forces acting on the left half of the support mechanism, the section method is employed, as shown in Figure 7. This portion can be simplified as a cantilever beam model. When the cantilever beam is subjected to an external force $F_{N}$, it undergoes bending deformation, and the bending moment varies with the position of the section. In this model, the maximum normal stress $\sigma_{\max}$ occurs at the section with the maximum bending moment, specifically at section m-m. Based on this analysis, the following conclusions can be drawn:(18)$$\sigma_{\max}=\frac{M_{\max}}{W}$$(19)$$M_{\max}=F_{N}L$$
where $M_{\max}$ is the maximum bending moment, *L* is the distance from the knee joint to the nearest connecting pillar, and *W* is the section modulus, which is related to the geometry of the section. If the section is a rectangle with height *h* and width *b*, then the following holds:(20)$$W=\frac{bh^{2}}{6}$$

From Equations (Equation 18)–(Equation 20), it follows that:(21)$$h=\sqrt{\frac{6F_{N}L}{b\sigma_{\max}}}$$

### 3.3. Analysis of Structure Characteristics and Motion Capabilities

The leg support mechanism consists of a support plate and a support flange. To explore the effect of the support plate and columns on the stiffness of the support mechanism, the leg model was simplified based on the kinematic analysis of the four-bar mechanism and the motion range of the lower leg. MATLAB R2022a was used to simulate the robot knee joint motion (Figure 8 shows the simplified simulation). In the figure, the green dots represent the possible placement positions of the support flange, and these data are stored in a database. The leg links and lower leg are modeled as rectangular prisms. During the motion of the links and lower leg, the green dots that are passed over are removed from the database. By iterating through the minimum starting angle α of the knee joint, the placement points for the support flange that do not interfere with the robot leg motion are identified. The lowest coordinate position, $U_{\max}$, is then determined.

Based on the above analysis of the mechanical performance of the support mechanism, Equation (Equation 21) is derived, which in turn leads to the relationship between the position of the support flange and the thickness of the support plate. The relationship between the position of the support flange and the thickness of the support plate is shown in Figure 9.(22)$$l_{4}=U_{\max}+L$$

From this, the following can be derived:(23)$$h=\sqrt{\frac{6F_{N}\left(l_{4}-U_{\max}\right)}{b\sigma_{\max}}}$$

After discussing the effect of the support flange position on the thickness of the support plate, the relationship between the minimum knee joint position and the robot’s motion capabilities can be further investigated. Let the mass of the support plate per unit thickness be $m_{0}$, then, the mass of a single support plate is as follows:(24)$$m_{1}=hm_{0}=m_{0}\sqrt{\frac{6F_{N}\left(l_{4}-U_{\max}\right)}{b\sigma_{\max}}}$$

The total mass of the robot is given by the following:(25)$$m_{all}=m_{2}+8m_{1}$$
where $m_{2}$ represents the mass of the motor, torso, and other non-support plate components.

Subsequently, the relationship between the robot knee joint motion range and its dynamic performance is analyzed from an energy perspective. To quantify the dynamic performance, the maximum jump height *H* achieved by the robot during jumping motion is taken as the index of its dynamic performance. Figure 10 illustrates the simplified model of the robot jump. Figure 10a shows the minimum bending angle of the robot knee joint, while Figure 10b depicts the moment when the robot leaves the ground during the jump. In this model, it is assumed that at the moment of takeoff, the robot thigh joint is perpendicular to the torso, and the robot foot is in contact with the ground, transmitting a force $F_{f}$. When the robot foot completely leaves the ground, the following can be obtained:(26)$$F_{f}\left(d_{2}-d_{1}\right)=\frac{1}{2}m_{all}gH$$
where *H* represents the maximum height the robot can jump, $d_{1}$ is the height of the robot from the ground when it reaches its lowest crouching position, and $d_{2}$ is the height of the robot from the ground at the moment it leaves the ground during the jump.

The value of $F_{f}$ is complex during robot motion, as it varies with the knee joint movement and is difficult to derive through mechanical analysis. Therefore, in this study, $F_{f}$ is simplified as a linear elastic force model as follows:(27)$$F_{f}=k\left(d_{2}-d_{1}\right)$$
where(28)$$d_{1}=l_{6}\sin\left(\alpha \right)$$(29)$$d_{2}=l_{5}+l_{6}\sin\left(\alpha \right)$$The relationship between the robot maximum jump height *H* and the minimum knee joint angle α can be derived.(30)$$H=\frac{k{\left(l_{5}+l_{6}\sin\left(\theta_{5}\right)-l_{6}\sin\left(\alpha \right)\right)}^{2}}{1/2m_{all}g}$$
where $l_{5}$ represents the length of the thigh and $l_{6}$ represents the length of the lower leg.

## 4. Optimization of the Four-Bar Linkage Mechanism

This optimization approach minimizes joint torque and velocity requirements by balancing the motor output torque and speed. We utilize motion-trajectory optimization to integrate the target trajectory into the mechanism design process. As a result, when the robot follows a specified trajectory, peaks in speed and torque occur at distinct joint positions. Considering the variation in transmission ratio with joint position during operation, we balance the structure demands for both peak speed and torque. This section thoroughly examines the process of optimizing link lengths to enhance the dynamic performance of the mechanism.

### 4.1. Variable Transmission Ratio

This section offers an in-depth analysis of the transmission ratio characteristics of the four-bar linkage mechanism to evaluate its potential and provide guidance for the design methodology outlined in the subsequent section [25]. Specifically, we isolate the driving principles of the mechanism to simplify the mechanism depicted in Figure 4a. We subsequently define the mechanical parameters necessary for optimization, as shown in Figure 4b, which demonstrates a four-bar linkage mechanism for driving knee joint motion. Additionally, we specify the initial mechanism parameters $l={\left(\left[\begin{array}{cccc} l_{1} & l_{2} & l_{3} & l_{4} \end{array}\right]\right)}^{T}$ and $\theta_{4}$, which describe the fundamental mechanical configuration, range of motion, and assembly-friendly features. By modifying the linkage lengths via the variable transmission ratio properties of the four-bar linkage, we can adeptly balance the requirements for speed and torque, resulting in the optimized mechanism parameters *l*.

Based on the kinematic analysis of the four-bar linkage presented in the previous section, when the motor imparts an angular velocity $\omega_{1}$ to the input link $l_{1}$, the angular velocity of the lower leg $l_{3}$ is denoted as $\omega_{3}$. The transmission ratio of the four-bar linkage mechanism in this scenario is defined as follows:(31)$$\eta_{a}=\frac{\omega_{3}}{\omega_{1}}$$
where $\omega_{1}$ is the angular velocity at the motor input, and $\omega_{3}$ is the angular velocity of the knee joint. According to the kinematic analysis in the previous section, the parameter that influences $\omega_{3}$ is *l*. Therefore, the parameter that affects the transmission ratio of the four-bar linkage mechanism is *l*. To study the effect of parameter *l*, linear increments Δl are added to $l_{1}$ and $l_{2}$ (since the mechanism is a parallelogram, changes in $l_{1}$ and $l_{2}$ result in the opposite changes in $l_{3}$ and $l_{4}$, which will not be discussed here). The value of Δl is set to 0.001. By simulating the motion of the leg mechanism, the knee joint angle and transmission ratio curves are obtained, as shown in Figure 11. The research results indicate the following: An increase in parameter $l_{1}$ will increase the transmission ratio. An increase in parameter $l_{2}$ will increase the transmission ratio within the knee joint angle range of 0–78°, while decreasing the transmission ratio in the range of 78–113°. Parameter $\theta_{4}$ determines the knee joint range of motion. Changing $\theta_{4}$ will either increase or decrease the knee joint angle, thus shifting the curve of the knee joint motion range and transmission ratio horizontally.

The results demonstrate that varying $l_{1}$, $l_{2}$, and $\theta_{4}$ yields distinct effects on the transmission ratio curve across different joint position ranges. This observation suggests that optimizing the initial parameters $l_{1}$, $l_{2}$, and $\theta_{4}$ can result in a more desirable transmission ratio curve.

### 4.2. Generation of Motion Trajectories

The jumping action of a quadruped robot entails the complex interaction between static and dynamic movements. Here, we utilize an optimal control approach to generate trajectories for the joint positions [26], torques, and velocities during the robot jumping phase, which serves as a foundation for optimizing the linkage length.The trajectory generation model is shown in Figure 12, illustrating the model coordinate system with the floating base coordinate system.

The generated jumping trajectory of the robot is shown in Figure 13. For clarity, only the motion trajectory of the hind leg knee joint is displayed, as the motion curves of both legs before and after the jump are similar.

Subsequently, the joint velocity and torque trajectories were analyzed in the joint position space. The joint position, joint velocity, and torque curves are shown in Figure 13. The joint space is intuitively divided into high-speed regions $\left[\omega_{h}\;\omega_{\max}\right]$ (yellow area in the figure) and high-torque regions $\left[T_{h}\;T_{\max}\right]$ (green area in the figure).

We use the motor T-N curve and power curve to define the high-speed point and the high-torque point (shown in the Figure 14). The red line represents the power curve, where the power reaches a maximum value at a certain speed $P_{\max}$ and then remains constant. The blue curve represents the motor T-N curve, where the torque starts from the maximum value and decreases with an increase in speed, and $T_{\max}$ is the maximum torque required for the action. We then define a high-speed–high-torque characteristic line, which is described by the following equation:(32)T=kω
where the horizontal coordinate of the point where it intersects the power curve is the high-speed point $\omega_{h}$. The vertical coordinate of the point where it intersects the T-N curve is defined as the high-torque point $T_{h}$. *k* takes the value $\left[0\;k_{\max}\right]$. The orange line is the characteristic line when *k* takes the maximum value $k_{\max}$. The quantitative impact of choosing k should be analyzed in future studies, where *k* takes the value of 1.08.

Figure 13 highlights the following trends: high-speed regions are predominantly observed at larger joint angles (approximately from 59° to 78°), whereas high-torque regions are concentrated at smaller joint angles (from 82° to 113°). By integrating these findings into the variable transmission ratio depicted in Figure 11, the optimal transmission ratio parameters were identified, as represented by the red curve in Figure 11.

## 5. Experimental Validation

### 5.1. Support Mechanism Verification

#### 5.1.1. Support Mechanism Simulation

The robot knee joint motion is simulated (Figure 15 shows the results when the minimum knee joint angle $\alpha =25^{\circ }$ and $\alpha =50^{\circ }$). The green dots indicate the possible placement positions of the support flange, and the red dots represent the lowest placement position of the support flange. Finally, the change curve of the lowest coordinate position, $U_{\max}$, of the support flange in the range from $\alpha =25^{\circ }$ to $\alpha =50^{\circ }$ is obtained, as shown in Figure 16. Detailed parameter values are provided in Table 1. The results show that as the minimum knee joint angle α increases, the lowest placement position of the support flange in the leg support mechanism moves downward.

#### 5.1.2. Comparison Experiment

Through the simulation described above, we have obtained $\alpha -U_{\max}$, which represents the relationship between the minimum angle of knee joint movement and the lowest position of the support flange. Here, $U_{\max}$ also corresponds to the optimal position of the support flange. Based on the relationships defined in Equations (Equation 23)–(Equation 25), we can derive the parameters for the leg support plate thickness (*h*), the mass ($m_{1}$), and the total mass of the robot ($m_{all}$). Subsequently, the dynamic performance of the robot is quantified by the robot jump height (*H*). The relationship between these parameters and the dynamic performance is established in Equation (Equation 33). The corresponding setup parameters are summarized in Table 1. The optimization process and results are shown in Figure 17.(33)$$\begin{array}{c} \begin{array}{cc} \max & H\;\;\;\mathrm{Equation}\;(\mathrm{30}) \end{array}\\ \begin{array}{cc} s.t. & \begin{array}{c} \mathrm{Optimal}\;\mathrm{support}\;\mathrm{flange}\;\mathrm{position}:\;\mathrm{Figure}\;16\;\;\\ \mathrm{Minimum}\;\mathrm{range}\;\mathrm{of}\;\mathrm{motion}\;\mathrm{of}\;\mathrm{the}\;\mathrm{knee}:\;\mathrm{Figure}\;16\;\;\\ \mathrm{Weight}\;\mathrm{of}\;\mathrm{support}\;\mathrm{plate}:\;\mathrm{Equation}\;(\mathrm{24})\\ \mathrm{Total}\;\mathrm{mass}\;\mathrm{of}\;\mathrm{robot}:\;\mathrm{Equation}\;(\mathrm{25}) \end{array} \end{array} \end{array}$$

The relationship between the robot jump height *H* and the achievable minimum knee joint angle α is shown in Figure 18. In the figure, the red curve represents the relationship derived from Equation (Equation 30). The blue dashed line represents the curve when the robot is kept with the mass of the support plate held constant at the initial design parameters.The dynamic performance of the robot is improved by about 7% after optimizing the structural parameters with the same range of motion as the knee joint.

In the design of the robot leg mechanism, the thickness of the leg support plate *h* and the minimum knee joint range of motion α can be determined based on the required motion capabilities. Table 2 and Table 3 present the structure design parameters for the maximum jump heights *H* of 0.65 m and 0.59 m, respectively. By adjusting these design parameters, the robot motion performance can be effectively improved to meet the specific task requirements.

### 5.2. Transmission Mechanism Experimental Validation

By comparing Figure 11 and Figure 13 with the optimized linkage lengths, the optimization results show that reducing $l_{1}$ and increasing $l_{2}$ successfully fit the transmission ratio curve to the target trajectory. In the high-torque region, the transmission ratio is increased to enable the mechanism to output a higher torque, while in the high-speed region, the transmission ratio is decreased to allow the mechanism to achieve a higher rotational speed. This approach effectively reduces both the peak motor torque and peak motor speed, alleviates the mass of the leg mechanism, and provides the necessary power support for the mechanism to perform higher intensity movements. By comparing the values of the reduction ratios in the high-torque and high-velocity regions before and after optimization, the peak torque requirement for the robot knee motors was reduced by 18.5% and the peak speed requirement for the motors was reduced by 24.8%. Furthermore, the optimized parameters are very similar to the original ones, which makes future adjustments to the mechanism more straightforward and convenient. The original and optimized parameter values are presented in Table 4.

To validate the simulation results, we conducted physical experiments using 3D-printed link structures both before and after optimization. To ensure the reliability of the experiment, the link structures were fabricated using 3D printing techniques in both cases. Given the challenges in accurately obtaining single-leg data due to the simultaneous motion of all four legs, we fixed the trunk of the quadruped robot and retained only one leg for testing. The motion was restricted to the hip and knee joint actuators to isolate the leg dynamics. Furthermore, to simulate the kicking motion of the leg during the robot jump, we inverted the robot and attached a weight equivalent to one-quarter of the total robot weight (2.5 kg) to the foot. The kicking velocity was increased to better replicate the dynamic conditions of the jumping motion. Finally, we repeated the leg kicking motion seven times and recorded the motor feedback data for both the pre-optimized and post-optimized structures, providing empirical validation for the simulation results.The experimental procedure is shown in Figure 19. The obtained data are shown in Figure 20.

In the experimental results depicted in Figure 20, the optimized structural parameters demonstrate a reduction in maximum joint torque of 17.7% and a decrease in maximum joint velocity of 19.8% compared to the original structure. While there is a discrepancy between the experimental and simulation results, these findings are sufficient to validate the accuracy of the simulation.

## 6. Conclusions

To enhance the dynamic performance of the legged robot, this paper presents an optimization method for leg mechanism parameters based on motion characteristics. Initially, a comprehensive mechanical analysis of the leg and an internal force analysis of the support mechanism are conducted to determine the motion range and performance requirements, which effectively guide the selection of support mechanism parameters. Subsequently, the variable transmission ratio characteristics of the bar linkage are combined with the robot motion characteristic curve to optimize the connecting bar linkage length. This optimization reduces the torque and velocity demands of the leg joint, achieving a reduction of 18.5% in the peak torque of the robot knee joint motor and a 24.8% decrease in the peak motor velocity compared to the original mechanism. The experimental results further validate the optimization method, demonstrating a 17.7% reduction in average maximum joint torque and a 19.8% reduction in average maximum joint velocity, despite some minor discrepancies between experimental and simulation results. Ultimately, this method successfully guides the design of a high-performance leg mechanism for the legged robot, providing a theoretical foundation for its stability and efficiency in complex environments. 

## Figures and Tables

**Figure 1 biomimetics-10-00173-f001:**
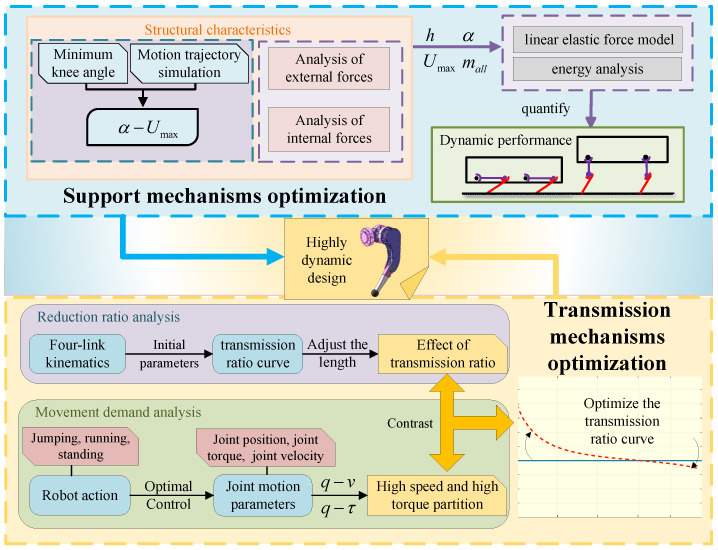
Overall framework of this paper.

**Figure 2 biomimetics-10-00173-f002:**
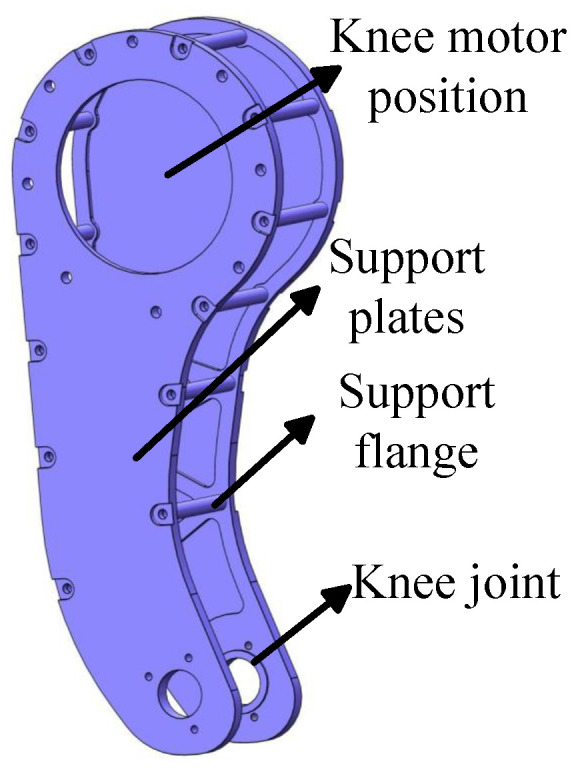
3D model of the supporting mechanism.

**Figure 3 biomimetics-10-00173-f003:**
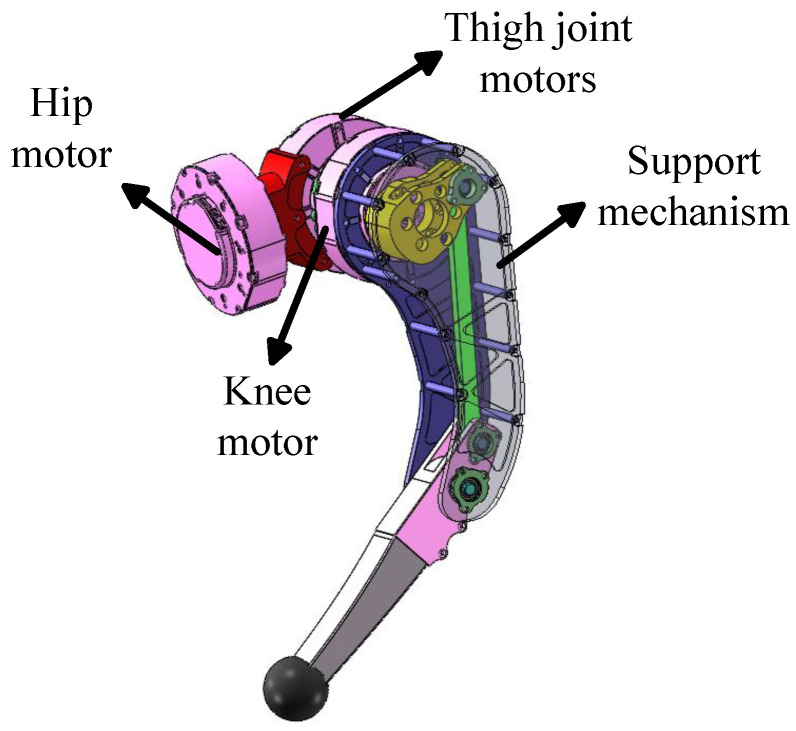
The model of the single-leg mechanism.

**Figure 4 biomimetics-10-00173-f004:**
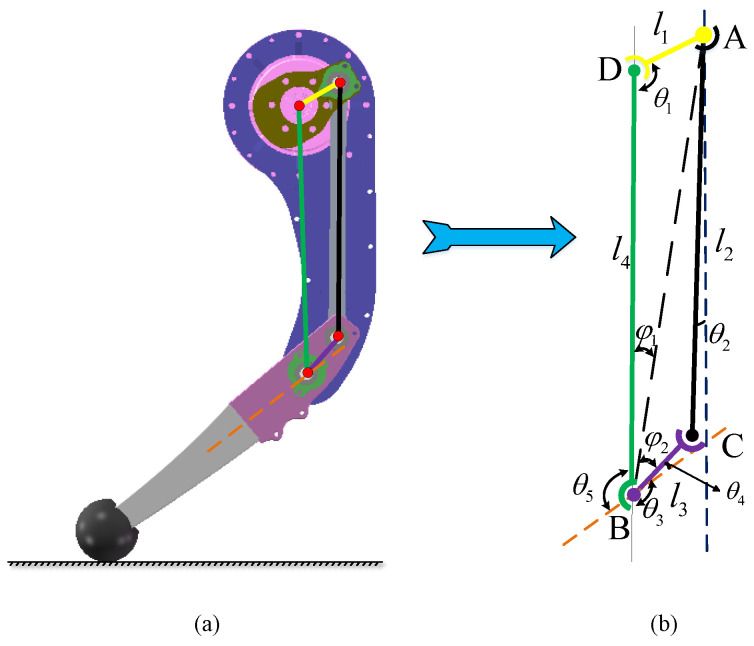
(**a**) Kinematic principles of the leg. (**b**) Principles of the four-bar linkage mechanism.

**Figure 5 biomimetics-10-00173-f005:**
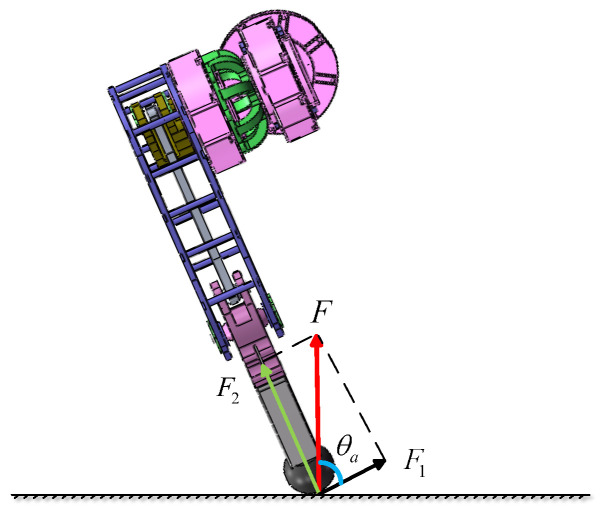
Force analysis of the leg.

**Figure 6 biomimetics-10-00173-f006:**
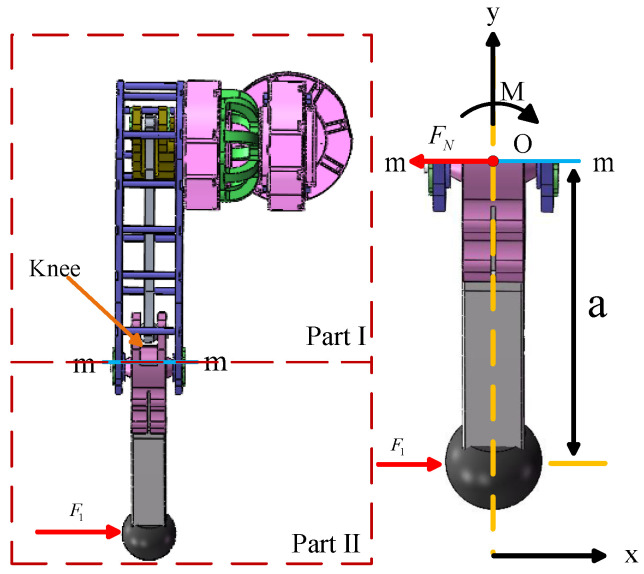
Internal force analysis of the leg.

**Figure 7 biomimetics-10-00173-f007:**
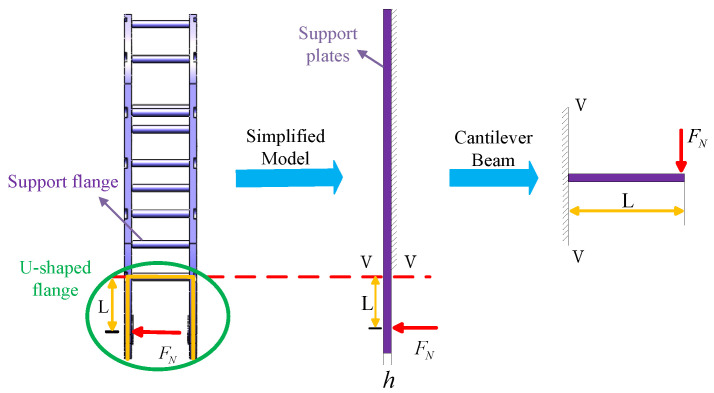
Force analysis of support plates.

**Figure 8 biomimetics-10-00173-f008:**
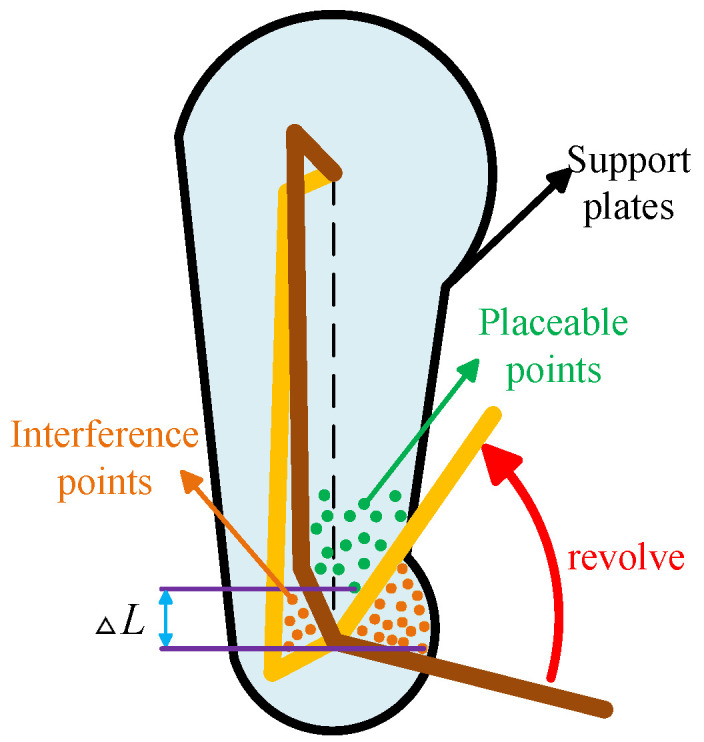
Simulation diagram.

**Figure 9 biomimetics-10-00173-f009:**
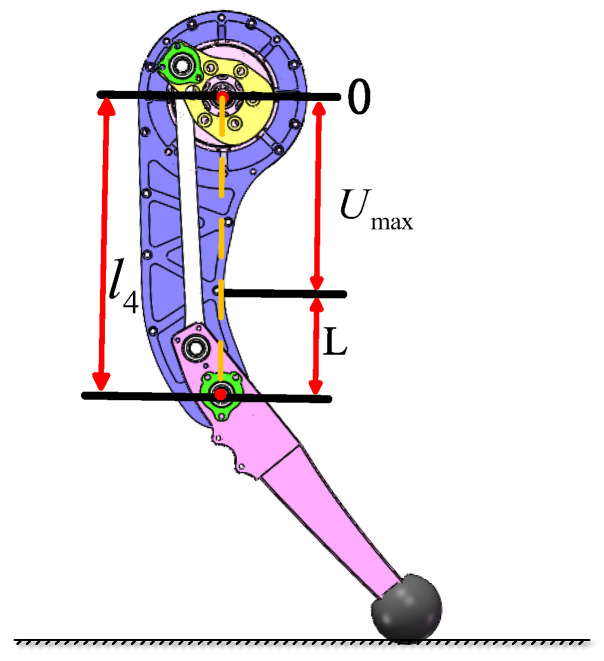
Position of the support flange.

**Figure 10 biomimetics-10-00173-f010:**
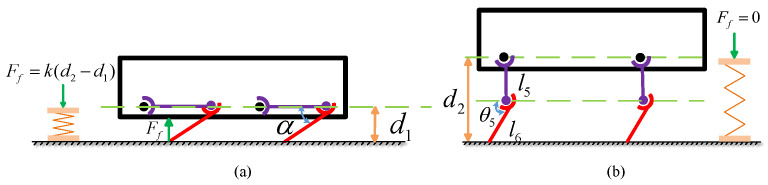
Jumping motion: (**a**) joint state at the minimum knee flexion angle during the jumping motion. (**b**) joint position state at the moment of takeoff.

**Figure 11 biomimetics-10-00173-f011:**
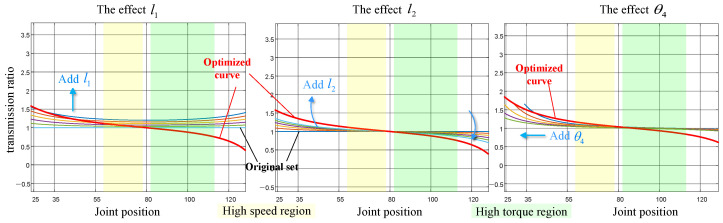
Optimized transmission ratio curve and the influence of various parameters on the transmission ratio.

**Figure 12 biomimetics-10-00173-f012:**
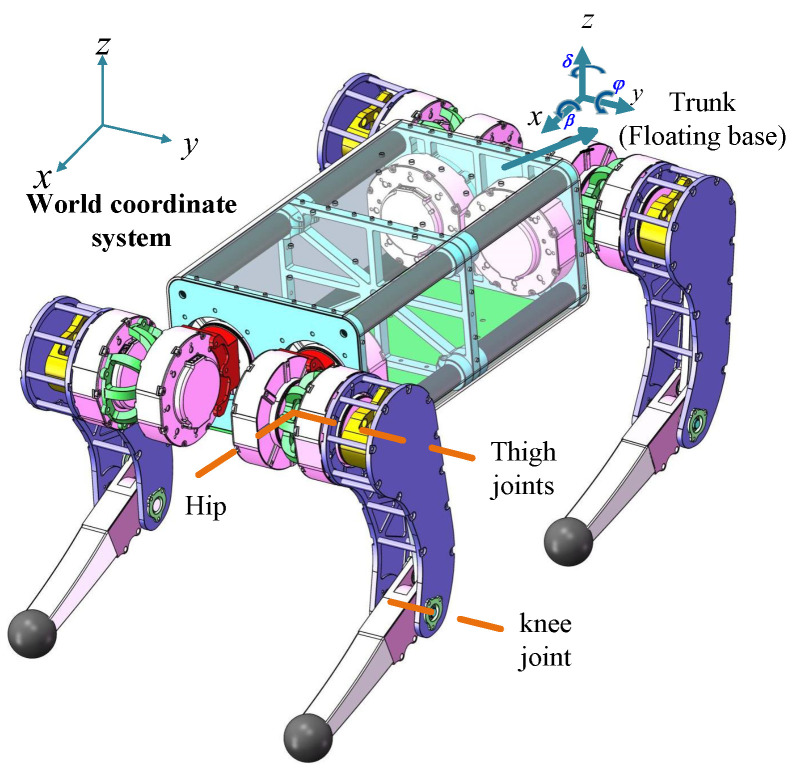
Joint linkage model of the quadruped robot. The trunk state is represented in the world coordinate system as $\left[x,y,z,\beta ,\varphi ,\delta \right]$, which corresponds to the floating base state.

**Figure 13 biomimetics-10-00173-f013:**
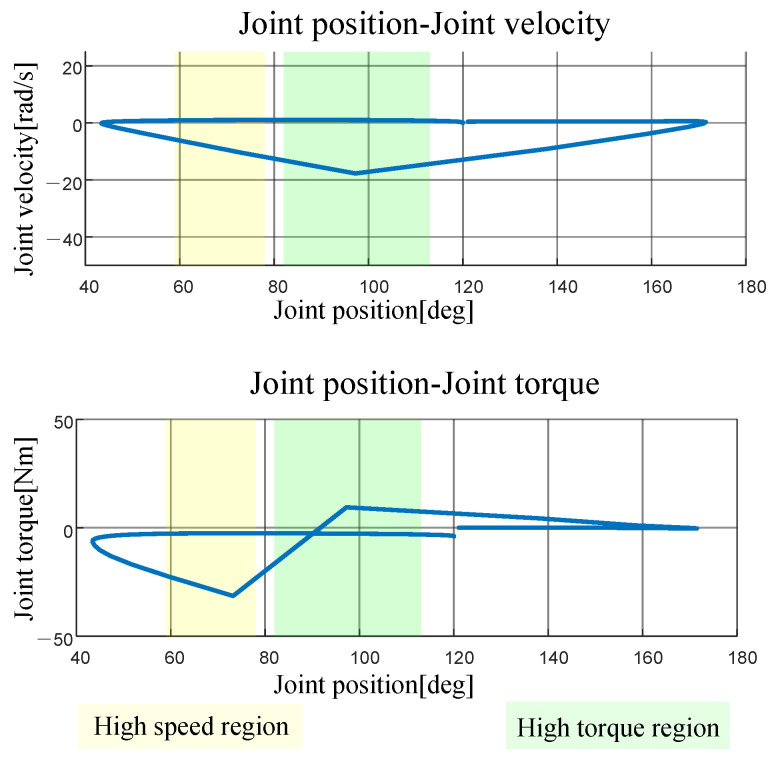
Joint position space trajectories of the knees.

**Figure 14 biomimetics-10-00173-f014:**
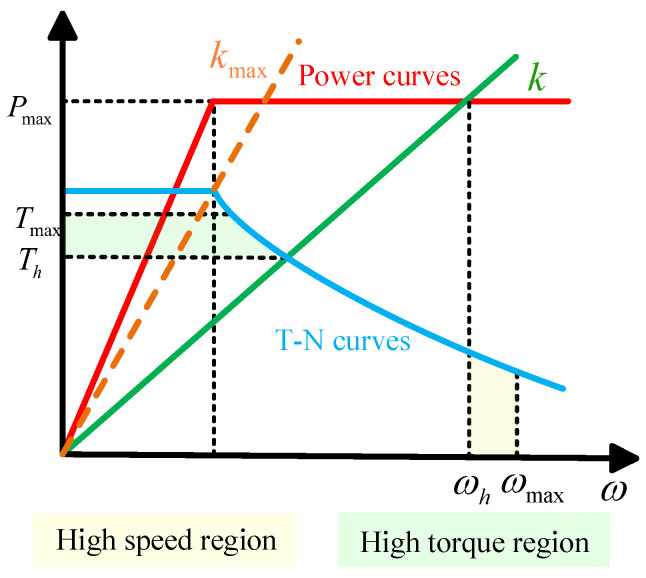
Motor characteristic curves.

**Figure 15 biomimetics-10-00173-f015:**
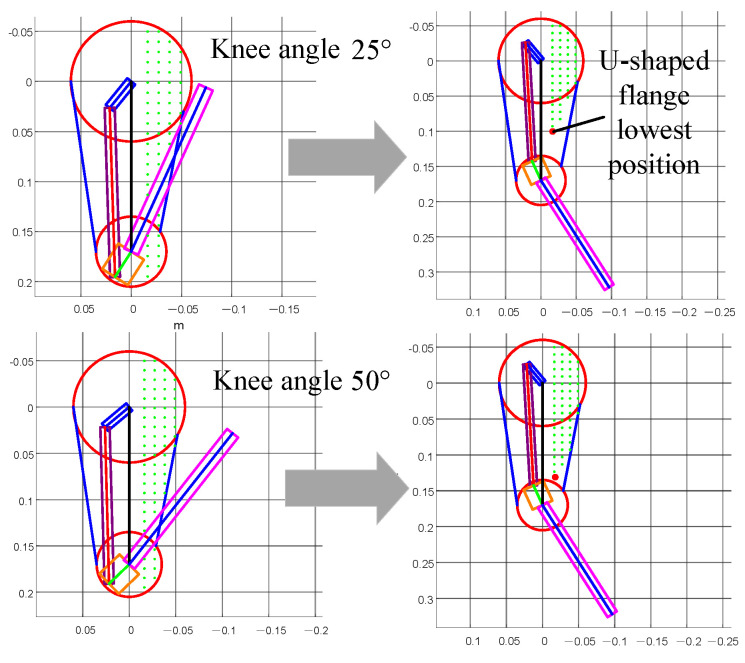
$\alpha =25^{\circ }$ and $\alpha =50^{\circ }$ simulation results.

**Figure 16 biomimetics-10-00173-f016:**
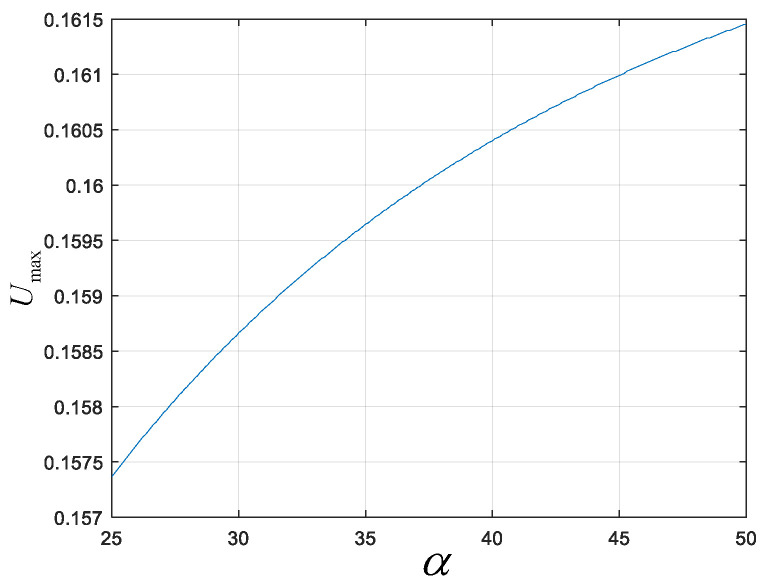
Variation curve of the lowest coordinate position $U_{\max}$ for the position of the support flange when the knee joint minimum angle α is in the range from 25° to 50°, as simulated.

**Figure 17 biomimetics-10-00173-f017:**
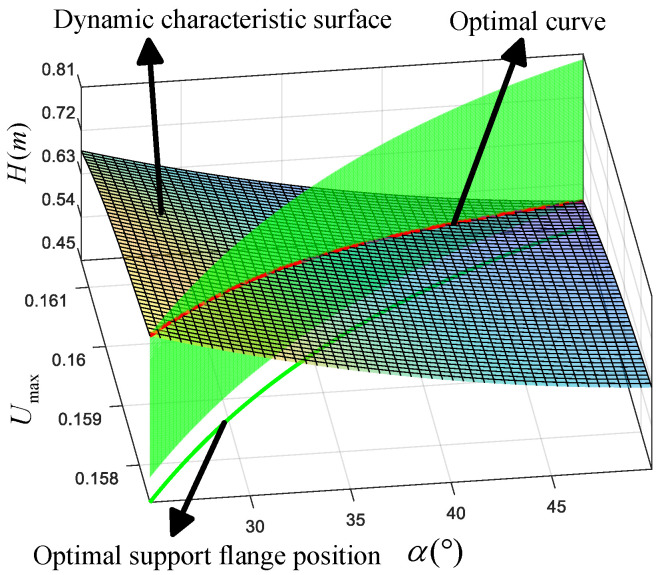
Dynamic performance under different parameters; the red line is the dynamic performance under the optimized optimal parameters.

**Figure 18 biomimetics-10-00173-f018:**
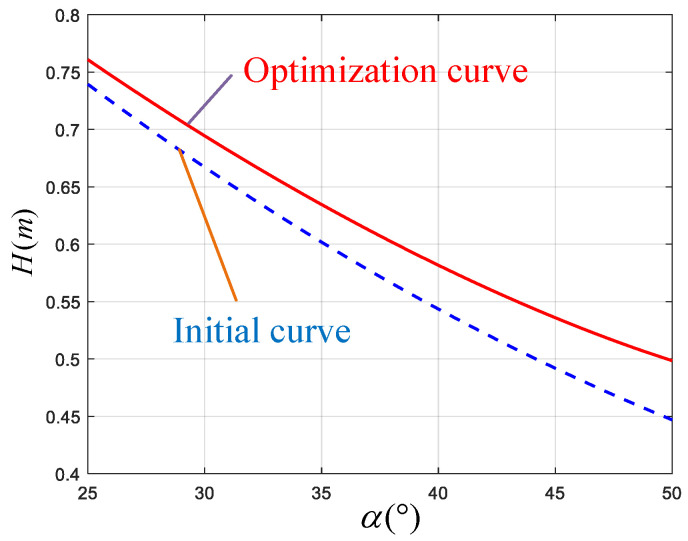
Curves illustrating the relationship between the minimum knee joint angle and the dynamic performance, comparing the initial and optimized parameters under quantified dynamic performance, as measured by the robot jump height *H*.

**Figure 19 biomimetics-10-00173-f019:**
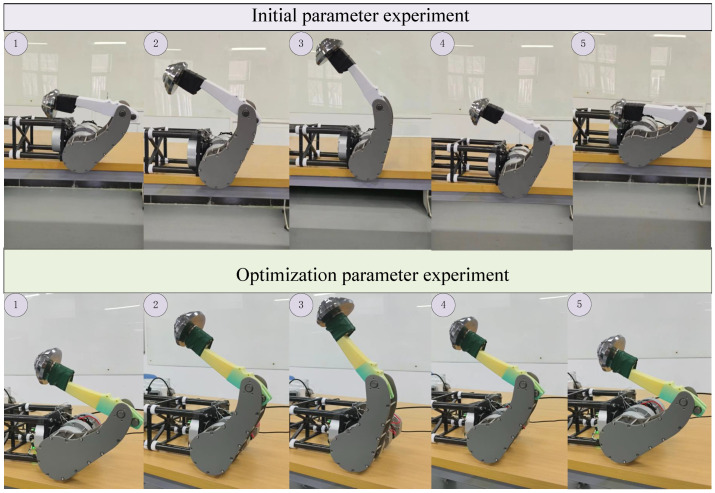
Experimental procedure of a single-leg inverted stance with a loaded foot during leg extension.

**Figure 20 biomimetics-10-00173-f020:**
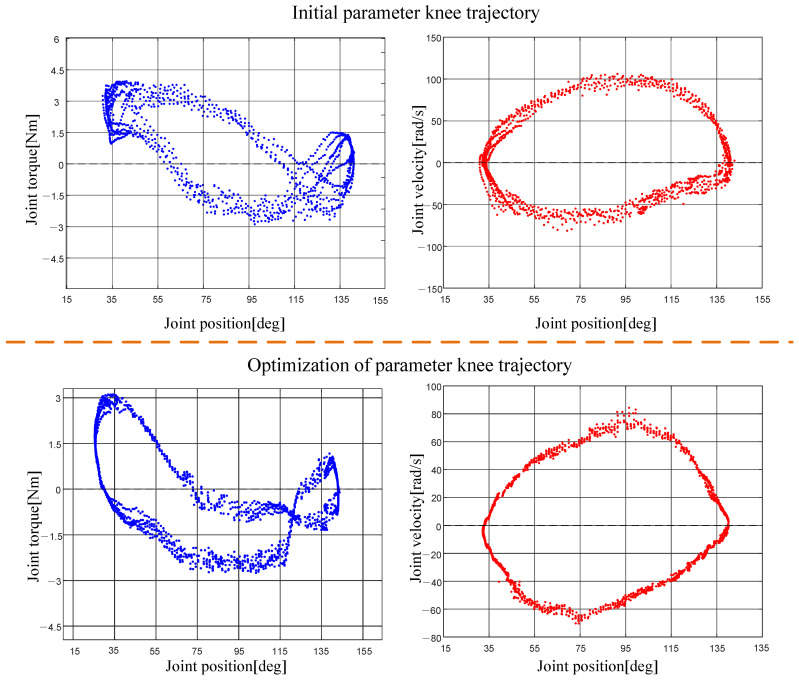
Experimental results of a single-leg inverted stance with a loaded foot during leg extension.

**Table 1 biomimetics-10-00173-t001:** Parameter Values.

Parameter	$F_{N}\left(N\right)$	b(m)	$\sigma_{4}\left(\mathrm{Mpa}\right)$	$\theta_{5}{(}^{\circ })$
Values	150	0.01	55.2	75

**Table 2 biomimetics-10-00173-t002:** Support mechanism design results for H=0.65.

*H*	α ${/}^{\circ }$	$U_{\max}$/m	*h*/m	$m_{1}$/kg
0.65	25	0.1574	0.0055	0.22

**Table 3 biomimetics-10-00173-t003:** Support mechanism design results for H=0.59.

*H*	α ${/}^{\circ }$	$U_{\max}$/m	*h*/m	$m_{1}$/kg
0.59	30	0.1586	0.005	0.02

**Table 4 biomimetics-10-00173-t004:** Linkage optimization results.

Parameter	Before Optimization	After Optimization
$l_{1}$ (m)	0.03 (m)	0.029 (m)
$l_{2}$ (m)	0.17 (m)	0.179 (m)
$l_{3}$ (m)	0.03 (m)	0.03 (m)
$l_{4}$ (m)	0.17 (m)	0.17 (m)
$\theta_{4}\;{(}^{\circ })$	0 ${(}^{\circ })$	9 ${(}^{\circ })$

## Data Availability

The original contributions presented in this study are included in the article. Further inquiries can be directed to the corresponding author.

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
