# Peer review of "Dynamic Motion-Based Optimization of Support and Transmission Mechanisms for Legged Robots"

_biomimetics, 2025, doi:10.3390/biomimetics10030173_

Round 1

Reviewer 1 Report

Comments and Suggestions for Authors

The authors of the article improve the dynamic characteristics of the knee joint mechanism of a four-legged robot. The ultimate goal is to reduce the peak values of the torque and speed during the jump. At the same time, the specified maximum jump height and the required stiffness of the mechanism are ensured. To solve the problem, authors developed mathematical and simulation models of the support and transmission mechanisms, the parameters of which are adjusted in the best possible way.

The provided description of the solution procedure allows to understand the general idea of the proposed approach. However, there is a lack of a clear formalization of the optimisation problem. Why are the obtained curves called optimal or optimized? What specific objective indicators were used as optimality criteria?

In addition, there are small comments and questions on the text and figures.

Section 3.1. In formula (13), the jump height is designated as X, and further in the text H is used.

Section 4.1. The grade "since the mechanism is a parallelogram..." is unclear. In Fig 4b it is not a parallelogram. And in Table 4 the "after optimisation" results show it. The caption to Fig. 11 should be corrected.

Section 4.2. Strange diagram of the joint position-velocity. Why does the position change at zero velocity (horizontal section of the blue line)?

Author Response

Comments 1:[The provided description of the solution procedure allows to understand the general idea of the proposed approach. However, there is a lack of a clear formalization of the optimisation problem. And a description of the method process is added above the formula.]

Response 1:[Thank you for your valuable feedback. We have added Equation (33) in the main text to more clearly describe the overall approach and the formalized expression of the optimization problem.]

Comments 2:[Why are the obtained curves called optimal or optimized? What specific objective indicators were used as optimality criteria?]

Response 2:[Thank you for your valuable feedback. We have further explained in detail the definition of the optimal curve and the optimality criterion. We have added the optimal curve obtained in this study (Figure 17). This curve is the optimal solution obtained by comprehensively considering the minimum angle of the knee joint, the position of the support flange and the jump height. Since the range of motion of the knee joint directly affects the position of the support flange, which in turn affects the thickness and mass of the support structure, and ultimately affects the dynamic performance of the robot. And through mechanical analysis, we evaluated the effect of the support flange on the stiffness of the support structure, thereby determining its impact on the overall structural quality. In addition, we also performed kinematic simulations to analyze the relationship between the range of motion of the knee joint and the position of the support flange. Finally, through energy analysis, the dynamic performance was quantified with the jump height as the evaluation index, so as to determine the optimal parameters.]

Comments 3:[    In addition, there are small comments and questions on the text and figures.
    Section 3.1. In formula (13), the jump height is designated as X, and further in the text H is used.
    Section 4.1. The grade "since the mechanism is a parallelogram..." is unclear. In Fig 4b it is not a parallelogram. And in Table 4 the "after optimisation" results show it. The caption to Fig. 11 should be corrected.]

Response 3:[Thank you for the remark. We have standardized the notation for jumping height throughout the paper by replacing all instances of \(X\) with 
\(H\) to ensure consistency. Additionally, we have removed the inaccurate description of the parallelogram in Section 4.1 to enhance the precision of the explanation. Furthermore, we have revised the title of Figure 11 for better clarity, renaming it to "Optimized transmission ratio curve and the influence of various parameters on the transmission ratio."]

Comments 4:[ Section 4.2. Strange diagram of the joint position-velocity. Why does the position change at zero velocity (horizontal section of the blue line)?] 

Response 4:[Thank you for the remark. Figure 13 illustrates the trajectory of the robot’s jump. In the joint position and velocity curves, there are no horizontal segments. After the robot takes off from the ground, the joint velocity becomes relatively small. However, in a figure where the maximum velocity reaches 19 rad/s, such small variations are difficult to discern. Nevertheless, since the unit of joint position in the figure is degrees, even a velocity change of 1 rad/s results in a position change of approximately 57.3 degrees per second, making the variations in joint position clearly observable.]

Reviewer 2 Report

Comments and Suggestions for Authors

Figure 3 should be improved. It is difficult for the reader to see distinguish the four-bar mechanism. Probably, changing the color of the link with l2 length could solve the problem.

Equations (6) and (7) are identical.

How did you get equation (8)? Instead of doing that, and computing all the angles mentioned above (equations 2 and 3), you simpler could get the kinematic equations (10) and (11), considering Theta i angles starting from the same axis, the vertical one.

Comments on the Quality of English Language

English Language should be revised.

Reviewer 3 Report

Comments and Suggestions for Authors

This paper proposed a method to optimize the parameters of leg mechanism of robots based on dynamic motion. The proposed method consists of two key parts: support mechanism optimization and transmission mechanism optimization. For the support mechanism, a mechanism analysis index based on robot motion energy was introduced to evaluate the robot dynamic motion performance. Under the structure stiffness constraint, the index was enabled to quantitatively analyze the influence of the range of motion and structure mass on the robot motion performance guiding the design of parameters such as the range of motion, structure thickness, and U-flange position of the mechanism. For the transmission mechanism, the paper optimized the linkage length and knee joint angle for transmission ratio. Considering the variable transmission ratio and robot motion characteristics, the parameters were optimized to reduce the torque and speed requirements of the leg joint. The proposed method determined the optimal mechanism parameters for dynamic performance based on specified motion energy requirements, and it also optimized the linkage length. The key results showed that the peak torque of the knee joint motor was reduced by 18.5%, and the peak speed was reduced by 24.8%.

The concepts presented possess high interest in the field. However, the paper needs to be improved before it can be published. The title should specify the name(s) of the mechanisms rather than simply using ‘mechanism optimization’. The paper is based on simulation studies. Verification and validation of the simulation results using physical robots are necessary to justify the proposed mechanisms because there are many constraints in real-world systems that are not properly reflected in simulation studies. Moreover, the simulation environment and procedures have not been presented clearly and adequately. The simulation results are also inadequate to justify the conclusions. The authors must introduce the novel optimization theories that they used to optimize the mechanisms. The theoretical aspects of the optimizations are not clear. It is not clear whether the authors proposed to apply two mechanisms independently (modularly) or conjugately. The authors should explain how those two mechanisms are complementary if they are applied together. The authors need to justify the novelty in their proposed mechanisms, as well as they should compare their results with the state-of-the-art results to justify their contributions.

As a whole, the theoretical aspects of the optimization methods including their novelty and the adequacy of assessment results to justify the mechanisms should be addressed properly.

Comments on the Quality of English Language

Can be improved. 

Author Response

Comments 1:[The concepts presented possess high interest in the field. However, the paper needs to be improved before it can be published. The title should specify the name(s) of the mechanisms rather than simply using ‘mechanism optimization’. ]

Response 1:[Thank you for your valuable feedback. We agree with your observation that the title should specify the names of the mechanisms involved, rather than using the general term ‘mechanism optimization’. Based on your suggestion, we have revised the title to more clearly reflect the focus of our work. The new title is “Optimization of Support and Transmission Mechanisms for Legged Robots Based on Dynamic Motion”, which specifies the two key mechanisms addressed in our paper.By optimizing the structural parameters of these two mechanisms, the dynamic performance of the robot is improved.

We believe this change enhances the clarity and specificity of the paper, and we thank you again for helping us improve the overall presentation.]

Comments 2:[The paper is based on simulation studies. Verification and validation of the simulation results using physical robots are necessary to justify the proposed mechanisms because there are many constraints in real-world systems that are not properly reflected in simulation studies.]

Response 2:[Thank you for your  remark. We have added physical experiments on the robot in Section 5.2. In these experiments, we fixed the trunk of the quadruped robot and retained only a single leg for testing, with motion restricted to the hip and knee joint actuators. Additionally, to simulate the kicking motion during the robot's takeoff, we inverted the robot and attached a weight equivalent to one-quarter of the robot’s total weight (2.5 kg) to the foot. To better replicate the dynamic conditions of the jumping motion, we increased the kicking velocity. Finally, we repeated the kicking motion seven times and recorded the motor feedback data for both the pre-optimized and post-optimized structures. Compared to the original structure, the optimized structure exhibited a 17.7\% reduction in peak joint torque and a 19.8\% reduction in peak joint velocity. The experimental results are presented in Figure 20.]

Comments 3:[Moreover, the simulation environment and procedures have not been presented clearly and adequately. The simulation results are also inadequate to justify the conclusions.  The authors must introduce the novel optimization theories that they used to optimize the mechanisms.

Response 3:[Thank you for your comments. For the support mechanism, we added the introduction of the method flow, and added formula (33) to clearly express the simulation process, and compared the optimized structural parameters with the original structural parameters, as shown in Figure 17. The results show that the dynamic performance of the optimized structural parameters is significantly higher than that of the original structure. For the transmission mechanism, a physical robot verification was added, and the experimental data is shown in Figure 20. The results show that the average maximum joint torque was reduced by 17.7\% and the average maximum joint speed was reduced by 19.8\%.]

Comments 4:[ The theoretical aspects of the optimizations are not clear. It is not clear whether the authors proposed to apply two mechanisms independently (modularly) or conjugately. The authors should explain how those two mechanisms are complementary if they are applied together.]

Response 4:[Thank you for your valuable feedback. The optimization of the transmission mechanism and the support mechanism are complementary, as both aim to enhance the dynamic performance of the robot . When the support mechanism parameters are fixed, the length of the fixed link in the four-bar linkage can be known, thereby guiding the design of the transmission mechanism. To clearly express the relationship between these two components, we have added an explanation of their relationship in the introduction section and modified the article framework diagram in Fig. 1 to better illustrate their connection.

Comments 5:[ The authors need to justify the novelty in their proposed mechanisms, as well as they should compare their results with the state-of-the-art results to justify their contributions. As a whole, the theoretical aspects of the optimization methods including their novelty and the adequacy of assessment results to justify the mechanisms should be addressed properly.]

Response 5:[Thank you for your valuable comments. Other researchers related research mainly focuses on the impact of various structures or drive modes on the dynamic performance of robots. However, no one has studied the position and range of motion of the U-shaped flange. In addition, due to the different structures, the comparison of optimization results is not very valuab In response to your suggestions, we have revised our manuscript to better highlight the contribution and innovation of our proposed method. Specifically, we have added Figure 17 to visually demonstrate the relationship between the key parameters and the dynamic performance, which provides a clear basis for selecting the optimal parameters. Additionally, we have incorporated an experimental section to validate the effectiveness of our proposed method and demonstrate its advantages over other advanced approaches. The experimental results corroborate the simulation outcomes, further confirming the improved dynamic performance of the U-shaped flange structure. These revisions effectively emphasize the targeted nature and practicality of our method in enhancing the performance of legged robots.]

Reviewer 4 Report

Comments and Suggestions for Authors

This paper provides an advanced approach to optimize both the support structure and the transmission mechanism of a legged robot for improved dynamic performance. By analyzing the robot’s bending stresses in the support plates and tuning a four-bar linkage for the knee joint, this paper aims to minimize motor loads during high-intensity motions without sacrificing structural stiffness.

A notable feature is the integration of support-plate stiffness analysis with a trajectory-driven linkage design. Many prior studies focus either on mass reduction or on torque-speed requirements in isolation, whereas this manuscript balances both simultaneously. As a result, the trade-offs among flange placement, plate thickness, and variable-transmission linkage geometry are examined holistically to enhance performance without excessive compromise to structural rigidity or motor efficiency.

Overall, the conclusions are consistent with the data presented, showing that adjusting linkage dimensions and support parameters can lower motor stress while maintaining essential motion ranges. The paper itself is also well-structured and clearly demonstrates the methodologies and results. I recommend publication of the paper. However, the following suggestions are provided to further strengthen the work:

  1. Further validation could strengthen confidence in the reported improvements, particularly the reductions in peak torque and speed. Hardware tests or more advanced simulations under varied load scenarios would help confirm these benefits.

  1. A concise comparative table summarizing existing methods would also highlight the novelty and direct contributions of this work.

Author Response

Comments 1:[Further validation could strengthen confidence in the reported improvements, particularly the reductions in peak torque and speed. Hardware tests or more advanced simulations under varied load scenarios would help confirm these benefits.]

Response 1:[Thank you for your valuable feedback. We have added real machine experimental verification for the peak torque speed reduction, as shown in Figure 19. In the experiment, our legs moved at different speeds, which is suitable for different loads. For this reason, we have verified it on the physical robot. Specifically, we simulated the kicking motion of the robot during a jump by inverting the robot and attaching a weight equivalent to one-fourth of the robot total weight (2.5 kg) to its feet. To better simulate the dynamic conditions of the jumping motion, we increased the speed of the kick. Finally, we repeated the kicking motion seven times and recorded the motor feedback data for the pre- and post-optimized structures. Compared to the original structure, the optimized structure showed a 17.7\% reduction in peak joint torque and a 19.8\% reduction in peak joint velocity.]

Comments 2:[A concise comparative table summarizing existing methods would also highlight the novelty and direct contributions of this work.]

Response 2:[Thank you for your suggestion. Other researchers related studies mainly focus on the effects of various structures or drive modes on the dynamic performance of robots, but no one has studied the position and range of motion of the U-flange. And no energy analysis was used to analyze the effect of the knee angle on high dynamic performance. For this reason, in addition to the study of the transmission mechanism, we also added the study of the support mechanism. In addition, due to the different structures, the comparison of the optimization results is not very meaningful. In response to your suggestions, we have revised the manuscript to better highlight the contribution and innovation of our proposed method. Specifically, we added Figure 17 to intuitively show the relationship between key parameters and dynamic performance, providing a clear basis for selecting the optimal parameters. In addition, we added an experimental section to verify the effectiveness of our proposed method and show its advantages over other advanced methods. The experimental results confirmed the simulation results and further confirmed the improvement of the dynamic performance of the U-flange structure. These modifications effectively emphasize the pertinence and practicality of our method in improving the performance of legged robots.]